# "Feeding the baby breast milk shouldn't be a problem" breastfeeding confidence and intention in pregnant persons with type 2 diabetes mellitus from Thailand

Ratchanok Phonyiam [1,2]*, Chiao-Hsin Teng[2,3], Yamnia I. Cortés[4], Catherine S. Sullivan[5], Aunchalee E. L. Palmquist[6], Eric A. Hodges[2], Marianne Baernholdt[7]

**1** Ramathibodi School of Nursing, Faculty of Medicine Ramathibodi Hospital, Mahidol University, Bangkok, Thailand, **2** School of Nursing, University of North Carolina at Chapel Hill, Chapel Hill, North Carolina, United States of America, **3** Chang Gung University School of Nursing, Taoyuan, Taiwan, **4** University of Iowa College of Nursing, Iowa City, Iowa, United States of America, **5** Carolina Global Breastfeeding Institute, Gillings School of Global Public Health, University of North Carolina at Chapel Hill, Chapel Hill, North Carolina, United States of America, **6** Duke Global Health Institute, Durham, North Carolina, United States of America, **7** University of Virginia School of Nursing, Charlottesville, Virginia, United States of America

* ratchanok.pha@mahidol.ac.th

## Abstract

Breastfeeding initiation has been found to be lower in pregnant persons with type 2 diabetes mellitus (T2DM). However, no studies have explored the potential impact of T2DM during pregnancy on breastfeeding plans among Thai pregnant persons. This study aimed to describe breastfeeding confidence and intention during pregnancy among Thai pregnant persons with T2DM. This qualitative analysis utilized data from a parent study with a convergent parallel mixed-methods design. This study was guided by the National Institute on Minority Health and Health Disparities (NIMHD) Framework. Eligible participants were pregnant persons diagnosed with T2DM, aged 20–44 years, and proficient in speaking Thai. The pregnant persons participated in semi-structured interviews and completed three questionnaires: demographic, infant feeding intentions, and breastfeeding self-efficacy. Data analysis involved descriptive statistics for quantitative data and directed content analysis for qualitative data. Twelve interviews revealed four main themes: breastfeeding intentions during pregnancy, breastfeeding confidence throughout pregnancy, breastfeeding barriers (such as previous challenging experiences and physical distance between mother and baby), and breastfeeding facilitators (including benefits and cost-effectiveness, consumption of Thai foods and herbs, and the availability of breast milk shipping services). This study offers insights into the intentions and confidence of Thai pregnant persons with T2DM regarding breastfeeding their baby after childbirth. To improve breastfeeding outcomes, the pregnancy period could serve as an opportunity to assess breastfeeding confidence, barriers, and facilitators that influence breastfeeding intentions among pregnant persons with diabetes.

**Data availability statement:** All relevant data are within the paper and its Supporting information files.

**Funding:** RP received the Alpha Alpha Chapter of Sigma Theta Tau International Research Grant, the Sigma Small Grant, and the Arthur C. Maimon Doctoral Student Research Award to complete this work. The specific grant/award numbers are not applicable. The funders had no role in study design, data collection and analysis, decision to publish, or preparation of the manuscript.

**Competing interests:** The authors have declared that no competing interests exist.

## Introduction

Since 2001, the World Health Organization (WHO) has recommended that infants should be exclusively breastfed for six months and to continue breastfeeding for two years or longer with appropriate complementary foods [1]. The World Health Assembly aims to achieve a global target of at least 70% exclusive breastfeeding (EBF) in the first 6 months by 2030 [1]. Breastfeeding involves providing human milk to an infant, while EBF is defined as providing only human milk without any other liquids or solids to the infant [2]. Currently, 44% of infants aged 0 to 6 months are exclusively breastfed worldwide [3]. Thailand has low breastfeeding rates [4]. In 2019, 34% of infants were breastfed within the first hour after birth; only 14% were exclusively breastfed during the first six months, down from 23% in 2016 [4].

Among Thai mothers, several potential factors are associated with their decision to start breastfeeding [5,6]. Breastfeeding attitudes and cultural factors about "being a good mother" were positively correlated with breastfeeding initiation in Thai mothers (p < 0.01) [7]; those with higher confidence scores intended to continue with exclusive breastfeeding for longer than persons with lower scores [6]. Persons with strong intentions are more likely to succeed at breastfeeding and willing to cope with unforeseen challenges [8]. In addition to these factors, breastfeeding rates may vary depending on a person's health complications during the course of pregnancy [6].

Moreover, among pregnant persons, those with diabetes showed an even lower rate of breastfeeding [9]. Diabetes is a common complication during pregnancy. Pre-gestational diabetes mellitus (PGDM) includes women with type 1 diabetes (T1DM: insulin deficiency) and type 2 diabetes (T2DM: insulin resistance) [10]. Gestational diabetes mellitus (GDM) is hyperglycemia during pregnancy [10]. High blood sugar levels delayed the onset of lactation and negatively impact breastfeeding plans [11]. Persons with PGDM are more likely to stop breastfeeding compared to those without diabetes before pregnancy [12]. A recent study [9] found that breastfeeding initiation rates were lower in persons with T2DM compared to those with T1DM, with predictors of initiation failure in T2DM including the intention to partially breastfeed.

With the onset of T2DM occurring at a younger age, Thailand is facing an increased prevalence of diabetes in children and adolescents [13]. A recent study of Thai persons younger than 30 years of age, showed that the average age at diagnosis is 20.8 years old. The prevalence of diabetes at younger ages (<30 years) is greater in females (60.2%) than in males (39.8%) [13]. The increasing prevalence of diabetes among younger women is particularly important as they contemplate and plan for pregnancy. The number of pregnant women with preexisting diabetes, rarely seen 30 years ago, has more than doubled from 0.7% to 1.5% of all pregnancies [14]. However, it is unknown how having T2DM in pregnancy may affect Thai pregnant persons' breastfeeding. Understanding how T2DM impacts breastfeeding initiation in Thailand is important because it can inform targeted interventions to improve breastfeeding rates. This knowledge is crucial for improving maternal and infant health outcomes in a country where breastfeeding rates are already suboptimal. Therefore, the purpose of this study was to describe breastfeeding confidence and intention in pregnancy among Thai pregnant persons with T2DM.

## Materials and methods

### Research design

This qualitative analysis utilized data from a parent study with a convergent parallel mixed-methods design [15]. The research primarily used a qualitative description approach [16]. Supplementary quantitative data was incorporated to provide sample characteristics for a better understanding of their breastfeeding plans. Our study protocol has been published elsewhere [17].

Our study was guided by the National Institute on Minority Health and Health Disparities (NIMHD) Framework [18]. This framework provides an understanding of sociocultural and behavioral domains of influence at individual, interpersonal, community, and societal levels [18]. The NIMHD framework was chosen because it considers health disparities across multiple domains (e.g., biological, behavioral, and sociocultural) and levels (i.e., individual, interpersonal, community, and societal) [18]. The framework aligns well with Thai culture, where the majority of people traditionally live in extended family units. This cultural fit makes it particularly relevant for understanding health and social dynamics within Thai society. Pregnant persons in Thai culture collaborate, learn, and share health practices, including diabetes management and breastfeeding, with other generations, including grandparents and relatives [19,20]. Theoretical framework integration spanned across the study design, data collection, data analysis procedures, and results presentation.

The Institutional Review Boards from two universities approved all the procedures, study materials, and personnel before study implementation (IRB 21-1477 from the United States and IRB 3428 from Thailand).

## Setting and relevant context

This study recruited pregnant persons with T2DM at a medical tertiary hospital in Bangkok, Thailand. As of 2023, Bangkok has an estimated population of approximately 10.5 million people. A recent national survey in Thailand reported the diabetes prevalence rate increased from 8.3% in 2004 to 10.8% in 2014 [21].

Thailand has experienced rapid social and economic changes, leading women to enter the workforce in greater numbers, from 44% in 1980 to 59.2% in 2019 [22]. Mothers who return to work are more likely to discontinue breastfeeding, and 31% of mothers send their infants to live with their grandmothers in other areas as they go back to work in Bangkok [22].

Our data collection site is a designated Baby-Friendly Hospital and has breastfeeding clinics where lactation support providers offer ongoing support to persons who breastfeed or encounter difficulties in breastfeeding. The hospital also promotes immediate skin-to-skin contact, encourages breastfeeding on demand, and assists mothers in initiating breastfeeding within the first hour after birth.

## Sample

Participants were recruited from the antenatal care clinic through purposive sampling, which targets information-rich cases that best contribute to the research questions based on eligibility criteria [23]. Potential participants were recruited through face-to-face outreach or phone calls by the research assistant (RA). Inclusion criteria included pregnant persons with T2DM, aged 20–44, and able to speak Thai. Exclusion criteria comprised those with life-threatening illnesses such as myocardial infarction or psychiatric conditions precluding safe study participation, given the potential for exacerbation and emotional distress. Participants were compensated with a $10 gift card (equivalent to 300 Baht) upon completion of data collection.

The sample size was determined by achieving data saturation whereby no new information emerged from data analysis, leading to the decision to stop recruiting participants [16].

## Measurement

Three questionnaires were included. First, the demographic questionnaire included a pregnant person's age, marital status, educational level, monthly household income, employment status, duration of diabetes, gravidity and parity, and gestational age.

Second, we used the Thai version of the 5-item Infant Feeding Intentions Scale (T-IFI; Cronbach's alpha = 0.857) to measure maternal intention to breastfeed their infant [24]. The T-IFI employs a 5-point Likert scale (0 to 4). The total score is derived by averaging the first two items and adding this average to the sum of scores for items 3–5. Scores range from 0 to 16, with 0 indicating a strong intention not to breastfeed and 16 indicating a strong intention to fully breastfeed as the sole source of nutrition until six months of age. There were no cut-off points for the T-IFI [25].

Third, we used the Thai version of the 14-item Breastfeeding Self-Efficacy Scale-Short Form (BSES-SF; Cronbach's alpha = 0.84) to measure confidence in breastfeeding their infant [26]. The BSES-SF employs a 5-point Likert scale (1 = not confident at all, 5 = very confident). Total scores range from 14 to 70, with higher scores reflecting greater confidence [27]. A cut-off of 50 was used: scores ≤50 indicated low breastfeeding confidence, while scores >50 indicated high confidence [28].

To address cultural bias, we utilized the Thai versions of IFI and BSES-SF. These scales have undergone cultural adaptation specifically for Thai people, ensuring relevance to breastfeeding intention and confidence in Thai context.

The interview guide, initially developed by the research team in English, included open-ended and follow-up questions on breastfeeding intention and confidence during their current pregnancy. It was translated into Thai for clarity and cultural appropriateness by two bilingual researchers. A pilot test with two pregnant persons with T2DM in Thailand ensured clarity (See S1 File).

## Data collection

The study was conducted from March 1 to October 31, 2022. Enrollment was conducted online through the Research Electronic Data Capture (REDCap) platform. Eligible participants scanned a QR code that linked to the electronic informed consent form. Upon completion of the consent form, participants proceeded to the survey using the same link.

The principal investigator (PI, RP) conducted semi-structured interviews, varying in length (10 to 49 minutes). Field notes on the interview date and tone were recorded. Audio recordings were de-identified and translated each interview transcript from Thai into English by a certified bilingual translator, with the PI (RP) then reviewing for cultural appropriateness and providing clarifications.

Participants were tracked using case identification (ID) numbers. Their names and telephone numbers were stored in a separate file. Interview data and transcribed interviews were stored on a secure server. Protection of participant's privacy and confidentiality was provided.

Qualitative data analysis occurred concurrently with data collection in an iterative process. English transcripts were analyzed using Atlas.ti version 9 (Atlas.ti Scientific Software GmBH, Berlin, Germany). Data analysis was based on directed content analysis where researchers conducted analysis with the predetermined codes derived from the NIMHD framework [29]. The NIMHD framework's sociocultural environment and behavioral domains across individual, interpersonal, and societal levels [18] were used as initial codes such as family functioning (interpersonal level) and policy and laws (societal level). Two coders, PI (RP) and non-Thai researcher (CHT), independently coded English interview transcripts, with a senior researcher (MB) serving as a third coder to resolve discrepancies. Any data that did not align with the NIMHD framework codes led to the creation of new codes.

The rigor and trustworthiness were achieved in four domains. First, to ensure credibility, we validated our findings through member checking [30]. We prepared a diagram with visual summaries, which allowed both a researcher and one participant (10 percent of enrolled participants) to review the relationships between themes and subthemes after all data collection

was completed [31]. We selected one participant who had recently given birth to review the results, as she could best recall her pregnancy experience. Second, dependability was ensured through consistent data collection procedures following our study protocol. Third, confirmability involved collaboration among three coders. Fourth, transferability was established by providing a detailed sample and setting description for comparisons with other contexts.

Quantitative data from the questionnaires were analyzed using IBM SPSS version 28.0, involving means, standard deviations, ranges for continuous variables, and frequencies and percentages for categorical variables. Imputation, substituting the mean value for missing data, was employed for handling missing data points [32].

### Inclusivity in global research

Additional information regarding the ethical, cultural, and scientific considerations specific to inclusivity in global research is included (see S1 Checklist).

## Results

### Sample characteristics

Twelve Thai pregnant individuals with T2DM, mean age 34.33 (Standard Deviation: SD = 4.29) years, participated. All participants were married, with 25% holding a bachelor's degree and another 25% holding a degree higher than a bachelor's. 41.7% worked in government or state enterprises. The mean monthly household income was 41,727.27 Baht (SD = 17,401.67). Gestational age ranged from 7 to 38 weeks (Mean = 21.42, SD = 13.03). Participants were diagnosed with T2DM between 3 weeks and 10 years prior (Mean = 45.42 months, SD = 41.68). Of 12 participants, seven previously gave birth (primipara/multipara), while five were pregnant for the first time (nullipara). See Table 1 and S1 Data for demographic details.

In Table 2, the T-IFI mean score was 10.41 (SD = 5.63), indicating that they have a higher intention to breastfeed the baby. The mean BSES-SF score was 45 (SD = 13.18). The participants were divided into two groups based on their BSES-SF total scores, with the mean score for the low-confidence group being 33.50 (SD = 11.94) and the mean score for the high-confidence group being 54.85 (SD = 3.00) (Table 2).

### Qualitative findings

Four main themes emerged including breastfeeding intention in pregnancy, breastfeeding confidence in pregnancy, breastfeeding barriers, and breastfeeding facilitators (Table 3).

### Theme 1: Breastfeeding intention in pregnancy

This theme identified three levels of influence (i.e., individual, interpersonal, and societal), with most findings falling within the individual level. Participants planned to breastfeed until their milk supply ran out, aiming for 3 months to 1 year, or until the baby stopped nursing. At the interpersonal level, participants planned to breastfeed until their baby stopped actively nursing. At the societal level, their intention to breastfeed was also influenced by maternity leave.

**Individual level.** Pregnant persons expressed their strong intentions to breastfeed prior to the birth of their infant. Some participants (4/12) indicated their intention to breastfeed their baby until they stopped producing breast milk, *"I would definitely feed the baby breast milk... Until I run out of breast milk"* (ID011). For participants who had more than one child, they shared their previous experience with their older child and intended to provide breastfeeding *"Until one year of age, like with my eldest child"* (ID004). Other participants (4/12) described

**Table 1. Participant characteristics (n = 12).**

| Variable | % (n) | Mean ± SD | Range |
|---|---|---|---|
| **Age (years)** | | 34.33 ± 4.29 | 27–40 |
| **Marital status** | | | |
| Marriage | 100% (12) | | |
| **Educational level** | | | |
| Secondary school | 25.0% (3) | | |
| Diploma | 16.7% (2) | | |
| Bachelor's degree | 25.0% (3) | | |
| Higher than bachelor's degree | 25.0% (3) | | |
| Others | 8.3% (1) | | |
| **Monthly Household Income (Baht)** | | 41,727.27 ± 17,401.67 | 10,000–70,000 |
| **Employment status** | | | |
| Government/State enterprise | 41.7% (5) | | |
| Merchant/Personal business | 8.3% (1) | | |
| Private company | 25.0% (3) | | |
| Freelance | 16.7% (2) | | |
| Others | 8.3% (1) | | |
| **Duration of diabetes (months)** | | 45.42 ± 41.68 | 3–120 |
| **Gestation** | | 21.42 ± 13.03 | 7–38 |
| **Gravidity** | | | |
| Primigravida | 8.3% (1) | | |
| Multigravida | 91.7% (11) | | |
| **Parity** | | | |
| Nullipara | 41.7% (5) | | |
| Primipara | 50.0% (6) | | |
| Multipara | 8.3% (1) | | |

SD, standard deviation.

**Table 2. Breastfeeding intention and breastfeeding confidence (n=12).**

| Variable and scale | % (n) | Mean ± SD | Range |
|---|---|---|---|
| Breastfeeding intention (T-IFI) | 100% (12) | 10.41±5.63 | 1.50–16 |
| Breastfeeding confidence (BSES-SF) | 100% (12) | 45±13.18 | 17–59 |
| •Low confidence (Total score ≤50) | 50% (6) | 33.50±11.94 | 17–50 |
| •High confidence (Total score >50) | 50% (6) | 54.83±3.00 | 51–59 |

SD, standard deviation.

an intention to breastfeed their baby for a certain period, which was ranging from three months until one year. One participant expressed uncertainty about the ability to feed a baby, *"I'm not sure whether I can feed my baby breast milk… If I can do it, then I want to do so"* (ID013).

**Interpersonal level.** Participants (2/12) reported their intention towards duration of breastfeeding depending on the interaction between a mother and a baby. One wanted to breastfeed and shared that *"Until the baby stops sucking on breast milk"* (ID009). Another participant also shared that *"I intend to feed my babies breast milk for six months… or until my baby doesn't want to suckle at my breast anymore"* (ID014).

**Table 3. Themes and subthemes.**

| Themes | Subthemes | | |
|---|---|---|---|
| | **Individual level** | **Interpersonal level** | **Societal level** |
| Breastfeeding intention in pregnancy | 1. Until running out of breast milk<br>2. Trying to feed babies breast milk for three months to one year | 1. Baby stopped actively nursing | 1. Breastfeeding duration determined by maternity leave |
| Breastfeeding confidence in pregnancy | 1. Diabetes has no effect on breastfeeding<br>2. Worrying due to diabetes during pregnancy | 1. Gaining information on diabetes in pregnancy from the internet and colleague | Not reported |
| Breastfeeding barriers | 1. Prior difficult experiences with breastfeeding such as insufficient breast milk supply and a baby's sucking issue | Not reported | 1. Physical distance between mother and baby makes it difficult to provide breastfeeding |
| Breastfeeding facilitators | 1. Breast milk is beneficial and cost-effective<br>2. Breastfeeding signifies a bond | 1. Support from husband and grandparent<br>2. Thai foods and herbs consumption and restrictions recommended over the generations | 1. Breastfeeding equipment such as a breast pump and a freezer facilitate breastfeeding<br>2. Breast milk shipping service |

**Societal level.** Some participants (3/12) explicitly stated their breastfeeding intention was determined by maternity leave. For example, one said *"Yes, I can take a 90-day maternity leave. I'll try to feed my baby breast milk… While I'm with my baby, I will be breastfeeding for the entire three months"* (ID007). Another participant planned to take a leave for six months with full salary for the first three months and half salary for the other three months, *"so that my child would be with me for as long as possible"* (ID006).

## Theme 2: Breastfeeding confidence in pregnancy

The theme identified two levels of influence, with most findings falling within the individual level. At the individual level, participants were concerned about the effect of T2DM on their breastmilk and whether it would impact their babies' health. At the interpersonal level, gaining information about diabetes in pregnancy from the internet and colleagues helped increase their confidence.

**Individual level.** Participants (11/12) expressed their confidence in breastfeeding, because they believed that diabetes has no effect on breastfeeding if they can control their blood sugar levels, *"I don't think it [diabetes] has any effect. This is my opinion. I haven't consulted the doctor about this...I think that if I eat according to the doctor's recommendations, I will be able to manage the blood sugar level during the breastfeeding period"* (ID004). However, one participant (1/12) expressed worry and uncertainty regarding whether having diabetes would negatively affect the quality of breast milk, *"I'm worried about whether there will be any effects on my baby if it is fed breast milk… I'm worried about the scenario that, when the time [breastfeeding] comes, I may not be able to take care of myself as well as I do now. My blood sugar level may not be stable due to hormones or something like that. I'm worried about whether there would be any effects on my baby if it were fed breast milk"* (ID009).

**Interpersonal level.** Participants (2/12) reported strong interpersonal support from a variety of sources. Most participants reported feeling confident as they read the information or comments from the internet and comments on social media from other expectant mothers who had diabetes during pregnancy and required insulin injections. One shared that *"On Google, I've just studied whether it would be safe for a person with diabetes to become pregnant"* (ID011). Another participant asked the colleague, who had diabetes during pregnancy, and the colleague said, *"Feeding the baby breast milk shouldn't be a problem. This senior colleague*

*at my school was pregnant and had to inject insulin as well. So, I sometimes talk to her"* (ID007).

## Theme 3: Breastfeeding barriers

Breastfeeding barriers were found to originate from both individual and societal levels. At the individual level, barriers to breastfeeding included previous difficult experiences. At the societal level, physical distance between mother and baby made breastfeeding challenging. We did not find participants reporting any breastfeeding barriers at the interpersonal level.

**Individual level.** A common individual-level barrier reported by participants (2/12) was a difficult experience in breastfeeding with a previous child, including issues like insufficient milk supply and problems with the baby's latch. For example, one participant (ID014) wanted to breastfeed a baby for six months but could not do it for any of them. Sometimes, they could only pump a small amount of breast milk, which was not enough for the baby (ID014).

**Societal level.** In Thai society, it is common for mothers to send their infants to be cared for by relatives in different provinces due to work or family obligations. This physical distance between participants (4/12) and the baby was reported as a significant barrier to breastfeeding, as it made it difficult for mothers to provide breast milk regularly. Participants had planned to send their babies to be cared for by family members after their three-month maternity leave ended. Two participants shared *"Because I'll send the baby to another province"* (ID001) and *"I send a baby to be cared for by grandma"* (ID006).

## Theme 4: Breastfeeding facilitators

The theme of breastfeeding facilitators identified three levels of influence: individual, interpersonal, and societal levels. The individual facilitators of breastfeeding included considerations of cost; the interpersonal facilitators were the significance of bonding, family support, the consumption of Thai food and herbs; the societal facilitator was the availability of resources.

**Individual level.** Two subthemes under individual facilitators 1) breast milk is beneficial and cost-effective and 2) breastfeeding signifies a bond. First, most participants (7/12) were aware of breastfeeding benefits and cost-effectiveness compared to infant formula. One shared that *"Breast milk is beneficial. Breast milk is more cost-effective than infant formula. Because I would only pay for the shipping"* (ID006). A participant stated that breastfeeding strengthened a baby's immune system: *"the longer the baby is fed breast milk, the stronger it will be"* (ID015). Second, many of the participants (5/12) shared that breastfeeding signifies a bond between mother and baby. When the participants breastfed the child, they felt a connection, which made them more willing to breastfeed. One said, *"There's a greater bond that comes from feeding the baby breast milk, especially if the baby suckles from my breast"* (ID013).

**Interpersonal level.** Participants who had the support of family members, such as husbands and grandparents, found it easier to breastfeed. They appreciated the assistance and encouragement they received. One participant said that a husband would warm up breast milk and feed the baby while I'm at work, making it convenient for them. One stated, *"He's willing to help so that the baby is fed breast milk"* (ID003).

In addition, several participants (4/12) in the study shared their beliefs that consuming traditional Thai foods and herbs such as Mai Nom Nang, ginger, and banana blossom in addition to prescribed medication can increase breast milk supply. These beliefs were passed down from generation to generation as family norms. For example, one participant's grandmother made her an herbal juice called Mai Nom Nang during her maternity leave, they also consumed banana blossom as well. Another participant consumed typical Thai foods that

were recommended for breastfeeding mothers, such as ginger and banana blossom, and did not take any medication to boost her milk supply (ID003).

There are also suggestions about food restrictions during breastfeeding. Participants indicated that they planned to restrict their diet even more than during their pregnancy period, including avoiding sweet and fatty foods because they believed that whatever mothers consumed will pass on to their babies. One participant noted, *"I believe that what I eat will get passed on to my child via my breast milk. So, I must restrict my diet"* (ID006). A participant restricted raw food and pickled food *"Because I'm not sure whether it's hygienic"* (ID006). Moreover, one participant reported that during the first pregnancy experienced clogged milk ducts caused by consuming too much fatty food because what they consumed would be passed on to their baby through breast milk. Therefore, they planned to restrict their diet more during breastfeeding.

**Societal level.** Participants (5/12) found it beneficial to utilize breastfeeding equipment, such as a breast pump and a freezer, to facilitate breastfeeding. One participant used a breast pump after returning to work, bringing it to pump milk at the workplace and then taking the milk home (ID013). A breast milk shipping service, estimated at 200–300 Baht per delivery, involved storing and transporting expressed milk from the mother's home to where the baby was staying. Participants preferred door-to-door services, like 'Inter Express,' over coach services, which required their relatives to pick up the expressed milk at a coach station (ID006).

## Member checking on qualitative findings

The PI (PR) conducted a phone meeting with an enrolled participant for member checking. The participant agreed with the finding. The participant added *"It depends on a mother's financial status and how much she has to spend on this equipment. I think the main factor for breastfeeding is whether the mom really wants to breastfeed a baby or not… that is the main factor and other factors may depend on the mother's status such as her job, income, or financial status"* (see S2 File).

## Discussion

To our knowledge, this is the first study to explore breastfeeding intention and confidence in Thai pregnant women with T2DM. The participants in our study exhibited a wide range of intentions and confidence levels regarding breastfeeding their baby after giving birth.

We found that having diabetes during pregnancy impacted participants' confidence to breastfeed more than their intention to do so. Our qualitative results aligned with a quantitative study that pregnant persons with T2DM in Australia are less likely to exclusively breastfeed compared to those without hyperglycemia during pregnancy [33]. Another study on persons with T1DM in Sweden found that those with diabetes may be more sensitive to disruptions due to their need for a structured routine in managing the condition, especially during breastfeeding [34]. Our results further explained that breastfeeding confidence was dictated by whether they would be able to manage their diabetes during pregnancy. Participants in our study were worried that hormonal changes would cause unstable blood sugar levels and unsure if feeding breast milk would be harmful to the babies. Persons with lower breastfeeding confidence tended to seek online information or consult colleagues with similar experiences regarding the impact of diabetes during pregnancy on their babies if they breastfed.

Participants noted that they had not discussed their breastfeeding plans with their providers including whether they could breastfeed their baby with T2DM. Our findings align with

a previous nationwide survey in Japan, which explored breastfeeding support for persons with GDM and identified barriers to its promotion [35]. Participants with GDM reported that breastfeeding support was lacking during pregnancy [35]. A lack of person-provider discussion may cause low health literacy on breastfeeding. Health literacy acts as a protective factor in maintaining exclusive breastfeeding and against early cessation during postpartum, as reported by 343 participants recruited from three hospitals in Spain [36]. In Thailand, mothers commonly visit lactation clinics and attend breastfeeding counseling [8]. Hospitals could improve accessibility to specialized breastfeeding clinics, offering ongoing care from providers. Tailored counseling can enhance health literacy and address concerns of pregnancy with T2DM.

For the T2DM group, it's important to collaborate with healthcare providers, like dietitians, to create a personalized meal plan that meets their health needs during breastfeeding. In this study, participants planned food restrictions, avoiding raw and fatty foods, believing it could affect breast milk composition. However, the CDC advises that breastfeeding individuals don't need to avoid specific foods [37], and previous research has shown that maternal food restrictions are unnecessary unless the baby shows a negative reaction to the food [38]. To prevent unnecessary dietary restrictions, providers should give accurate information on how diet impacts breast milk and the baby's health. This improved awareness helps support breastfeeding goals and ensures the well-being of both mother and child.

Participants acknowledged the use of traditional Thai foods and herbs, which have been known to increase or reduce breast milk supply. Certain foods such as banana blossoms help stimulate breast milk production [39] while ginger acts as a natural galactagogue for increasing the amount of breast milk [40]. Our study found it as a recommended food for breast milk production across generations which was surprising as it had not previously been stated in Thai literature [39,41]. Research may further explore the effectiveness of Thai traditional foods on breast milk production particularly in pregnant persons with T2DM.

Some participants expressed their intention to breastfeed their baby until they exhaust their breast milk, mirroring findings from a previous study involving mothers without diabetes residing in Bangkok [42]. Our study revealed that working mothers' breastfeeding duration was influenced by their three-month maternity leave. Freelancers, with more flexible schedules, tend to breastfeed for at least 6 months to a year. This aligned with past findings linking pregnant person's breastfeeding intentions to available nursing time, particularly for working mothers considering their return to work [8,43]. To increase the rate and length of breastfeeding, our findings underlined the need for maternity leave extension to six months from the current three months in Thailand [41]. In the Southeast Asian region, Vietnam stands out as a success story for having extended paid maternity leave from four to six months since 2013 which helped increase breastfeeding rate and duration [44]. Our study highlighted the importance of workplace policies and flexible work arrangements to support breastfeeding duration for working mothers.

For multiparous persons, their prior breastfeeding experience may influence their current breastfeeding intention. These findings were similar to other studies [45,46] in that prior breastfeeding experience may predict subsequent breastfeeding plans. Persons who did not breastfeed or encountered difficulties breastfeeding their older child were unlikely to express an intention to breastfeed their subsequent child [45,46]. We suggest that future longitudinal research measures maternal intention and confidence before their first birth, during breastfeeding, and before their next birth.

In Thailand, the physical distance between mother and baby makes it difficult to breastfeed. Internal migration increases the tendency for informal family separation [47]. The rural-to-urban flows for better health, work, and education have been increasingly observed in

Bangkok [47]. Parents often work in Bangkok while entrusting baby care to grandparents in distant areas, aligning with a study on Thai factory-working mothers [22]. Grandmothers can be key supporters of breastfeeding. Further research should explore their role in promoting breastfeeding [22]. In addition, participants recognized breast milk's benefits and cost-effectiveness, with breastfeeding shipping services helping maintain breastfeeding despite separation. Integrating milk shipping into prenatal education could reduce reliance on formula and support family planning [22].

## Limitations

This study has some limitations. Participants were recruited from a single outpatient antenatal clinic of a hospital. This study's participant bias may limit transferability to the broader Thai population, considering potential differences among clinics. Further research should broaden sampling across multiple hospitals for varied experiences. Variations in gestational age among participants may affect the study's comprehensive reflection on pregnancy. The potential overlap between T2DM and general breastfeeding barriers calls for cautious interpretation and future research to specify T2DM's unique impact on breastfeeding.

## Conclusions

This study highlights breastfeeding intentions, confidence, barriers, and facilitators of Thai pregnant persons with T2DM. Pregnancy serves as a crucial opportunity to screen and inform breastfeeding plans, emphasizing the need for education on diabetes's impact. Workplace policies and extended maternity leave can support working mothers. Nurses and lactation support providers can educate on traditional Thai foods affecting milk supply and offer personalized and culturally sensitive breastfeeding counseling for pregnant persons with T2DM.

## Supporting information

**S1 File. Interview guide.**
(DOCX)

**S2 File. Member checking.**
(DOCX)

**S1 Checklist. Inclusivity in global research.**
(DOCX)

**S1 Data. Dataset.**
(DOCX)

## Acknowledgments

The authors would like to acknowledge Ms. Pattaraporn Koonmee for her expertise and assistance in data collection in Thailand. We also thank Drs. Sangthong Terathongkum and Jiraporn Lininger for serving as the gatekeepers at the hospital. The authors would like to acknowledge Dr. Jittima Manonai Bartlett for her expertise in providing a consultant on the protocol of obstetrics and gynecology at the hospital level and health care system level. We are grateful to pregnant persons participating in our study and health care providers at antenatal clinics for support during our data collection. Finally, the authors thank Paul Mihas from the Odum Institute for Research in Social Science at the University of North Carolina at Chapel Hill for his invaluable expertise in providing consultation on the mixed methods approach.

## Author contributions

**Conceptualization:** Ratchanok Phonyiam, Chiao-Hsin Teng, Yamnia I. Cortés, Catherine S. Sullivan, Aunchalee E. L. Palmquist, Eric A. Hodges, Marianne Baernholdt.

**Formal analysis:** Ratchanok Phonyiam, Chiao-Hsin Teng, Marianne Baernholdt.

**Funding acquisition:** Ratchanok Phonyiam.

**Investigation:** Ratchanok Phonyiam.

**Methodology:** Ratchanok Phonyiam, Chiao-Hsin Teng, Yamnia I. Cortés, Catherine S. Sullivan, Aunchalee E. L. Palmquist, Eric A. Hodges, Marianne Baernholdt.

**Project administration:** Ratchanok Phonyiam.

**Supervision:** Yamnia I. Cortés, Catherine S. Sullivan, Aunchalee E. L. Palmquist, Eric A. Hodges, Marianne Baernholdt.

**Validation:** Ratchanok Phonyiam, Chiao-Hsin Teng.

**Writing – original draft:** Ratchanok Phonyiam.

**Writing – review & editing:** Ratchanok Phonyiam, Chiao-Hsin Teng, Yamnia I. Cortés, Catherine S. Sullivan, Aunchalee E. L. Palmquist, Eric A. Hodges, Marianne Baernholdt.

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
