## [Decision Letter · Decision Letter 0]

4 Sep 2024

PGPH-D-24-01048

"I’m Not Sure Whether I Can Feed My Baby Breast Milk" Experiences of Pregnant Women with Type 2 Diabetes Mellitus from Thailand

Dear Dr. Phonyiam,

Thank you for submitting your manuscript to PLOS Global Public Health. After careful consideration, we feel that it has merit but does not fully meet PLOS Global Public Health’s publication criteria as it currently stands. Therefore, we invite you to submit a revised version of the manuscript that addresses the points raised during the review process.

he manuscript has been evaluated by two reviewers, and their comments are available below. The reviewers have raised a number of concerns that need attention. In particular, they request revisions to improve the quality of the reporting in both the Methods and the Results, and revisions to improve the contextualization of the study.

Could you please revise the manuscript to carefully address the concerns raised?

We look forward to receiving your revised manuscript.

Kind regards,

Marianne Clemence

Staff Editor

Journal Requirements:

Additional Editor Comments (if provided):

Reviewers' comments:

Reviewer's Responses to Questions

**Comments to the Author**

1. Does this manuscript meet PLOS Global Public Health’s publication criteria ? Is the manuscript technically sound, and do the data support the conclusions? The manuscript must describe methodologically and ethically rigorous research with conclusions that are appropriately drawn based on the data presented.

Reviewer #1: Yes

Reviewer #2: Partly

2. Has the statistical analysis been performed appropriately and rigorously?

Reviewer #1: Yes

Reviewer #2: Yes

3. Have the authors made all data underlying the findings in their manuscript fully available (please refer to the Data Availability Statement at the start of the manuscript PDF file)?

Reviewer #1: No

Reviewer #2: No

4. Is the manuscript presented in an intelligible fashion and written in standard English?

Reviewer #1: Yes

Reviewer #2: Yes

5. Review Comments to the Author

Reviewer #1: This study described breastfeeding confidence and intention in Thai women with T2DM. The study employed semi-structured interviews, quantitative data collection, and directed content analysis using the NIMHD as a guiding framework to develop codes and themes. The findings on breastfeeding intentions, confidence, barriers and facilitators at individual, interpersonal and societal levels are valuable contributions to our understanding of breastfeeding confidence and intention in Thai pregnant persons with T2DM.

The strengths of this study include:

1. The incorporation of an appropriate theoretical framework to guide analysis and the examination of themes across different levels of influence

2. The establishment of rigor/trustworthiness of findings by member-checking and other methods

3. The concurrent presentation of both quantitative and qualitative data

I recommend several areas for major revision to improve the manuscript as a whole. These suggested revisions are mostly related to providing additional detail in the Methods and Results section, improving clarity of language, providing additional literature support in the Discussion section, among other items. Please also include continuous line numbers in the manuscript file upon resubmission.

1. The NIMHD framework is an important part of the study design/analysis, yet its description in the Methods section needs more detail. What was the rationale for using this framework? How did you employ this framework (e.g., a reader can figure out later on that you used the framework to divide themes, so perhaps you can state that in the Methods section)?

2. It is not clear how the authors decided on the question to focus on for member checking, or why only one participant was selected— were other participants asked to partake in the member-checking? Some additional detail on this in the Methods section would be valuable.

3. The short descriptions of themes (in the Results/Themes section) need additional detail (not more than 1-2 sentences each, though) to clarify what the themes are illustrating/what participants expressed in relation to the theme. For example, under theme 4 (Breastfeeding Facilitators”), I suggest rephrasing the sentence to “Facilitators to breasteeding included considerations of cost, family support and resource availability” or something similar so the reader understands the overarching message of the theme.

4. The inclusion of dietary considerations/eating of traditional Thai foods under the Interpersonal level of the Breastfeeding Facilitators theme seemed underexplained to me. What is the rationale for considering this as Interpersonal? Is it because food beliefs are considered peer/family norms? It may be helpful to return to the NIMHD framework for this section.

5. Are there any additional details about the study setting that can be included?

6. The use of gender-neutral pronouns is not consistent throughout. Thether the authors decide to use gender-neutral pronouns or pronouns based on participants’ identified gender (if collected during the study) is up to them, as long as it is consistent. I recommend the same for the term “pregnant persons” — this is used in the manuscript, but the term “women” is used in the abstract.

7. The Discussion section would benefit from additional literature to support the study’s findings and place them in dialogue with exisitng knowledge. For example, the sentence stating “previous literature found pregnant persons with T2DM were more likely to not exclusively breastfeed [33]” would benefit from additional detail about that study’s setting, the results and other literature supporting this finding.

8. Under Theme 2, it is stated that participants were worried about the effect of T2DM on milk, but then it is stated that all but one participant expressed their confidence in breastfeeding. This seems like an inconsistency between the conclusion from the data and the actual findings.

9. Tables should be labelled as Table S1, S2, etc since they are all supplementary files. Captions should be formatted as "Table S1 Text. _______". Please double-check the PLOS Global Public Health guidelines for all tables.

10. In the Discussion, page 17, the phrase “This cultural awareness” in regards to healthcare providers offering information on diet’s impact on breast milk seems a little awkward to me. Is “cultural” the right word? Perhaps reconsider, and use a term like “improved awareness.”

11. There are some places where sentences are not clear (possibly due to typos) or are in need of transitions/relocating— I would be sure to check for typos throughout. Here are the examples I found, by page number:

1. Page 4: “were found to be significantly positive correlations” — there is a word missing here

2. Page 6: Move the sentence about the gift cards to the paragraph above.

3. Page 11:

1. “On Google. I’ve just studied…” — do you mean, “On Google, I’ve just studied…”?

2. “We did not find interpersonal barriers.” Perhaps you could rephrase this as “We did not find that participants reported breastfeeding barriers at the interpersonal levell”

4. Page 13 — “and they also consumed banana blossom” — please change to “they consumed banana blossom as well.”

5. Page 15: “Longitudinal research measuring maternal intention during breastfeeding and after childbirth” is an incomplete sentence.

6. Page 16: Participants did not share that breastfeeding shipping costs as a barrier” — there seems to be a word missing here

Thank you for the opportunity to review this important work.

Reviewer #2: Key Points:

The paper shows a lot of promise and is very interesting! My biggest concern is the lack of emphasis on T2DM throughout the paper. More needs to be added into the paper in all areas (background, results, and discussion) on the experience of T2DM particularly, so that the paper focuses on how having diabetes impacts this sample’s intention to breastfeed and confidence in their ability to breastfeed in the future. I think it would be especially interesting to focus more on how many of the participants did not discuss it with their providers despite having concerns. It was mentioned briefly in your discussion but not in your findings. More details on this – and other examples of barriers specific to T2DM rather than general pregnancy/motherhood – would enrich your paper.

Title

- The non-quotation portion of your title should include the topic of breastfeeding that ties into your paper. At the moment it is very ambiguous and does not focus on the main issues that your paper addresses. For example: “Breastfeeding plans/intentions of pregnant women…”

Abstract

- The fourth sentence needs restructuring for ease of reading. For example: “This qualitative analysis utilized data from a parent study with…”

- The final sentence needs a comma between ‘confidence’ and ‘barriers’, rather than an ‘and’

Introduction

Overall:

- More information on Thailand would be useful to contextualize: pregnancy care, rate of T2DM in the population, how T2DM is cared for in the population, current breastfeeding rates (eg. initiation, EBF rates, any BF rates, how long they typically BF for), and why increasing breastfeeding rates would be beneficial.

- Why is this question important? You need to include more on why knowing the answer to how T2DM impacts breastfeeding initiation in Thailand is of interest to the reader.

First Paragraph:

- Remove the word ‘universally’ from your first sentence to help with flow of reading.

- You discuss EBF in the first 6 months. Does this mean ANY duration of time EBF or (what I think you are referencing) still EBF by 6 months? There is a big difference between initiation of EBF and then continuation up to 6 months in the statistics – especially in reference to your 2nd, 4th, and 5th sentences in that first paragraph – so be sure to clarify more specifically what those numbers refer to.

Second Paragraph:

- In the second sentence, it should either be “found to be significantly correlated…” or “found to have…”

Third Paragraph:

- Define briefly the three types of diabetes that you mention. Then give a more in-depth explanation of T2DM as this is the one that you will focus on in your paper.

- Why does having T2DM impact breastfeeding initiation? Add some more details on this.

Material and Methods

Overall:

- Instead of saying the position (eg. PI, non-Thai researcher, etc), put the initials of the person in brackets next to their role.

Research Design:

- Same comment as in the abstract that the first paragraph can be rephrased to help ease of reading.

- If you used both qualitative and quantitative then wouldn’t your study be mixed-methods instead? Especially if you used the T-IFI and BSES-SF which both have Likert scales for outcomes.

Setting and Relevant Context:

- More information would be useful here. What is the population in Bangkok? What does it mean to be designated Baby-Friendly for breastfeeding rates/outcomes (eg. encourages skin-to-skin, no formula advertised, mandatory BF training, etc)?

Sample:

- Define what purposive sampling means.

- In the second sentence, remove the word “comprised”.

Measurement:

- Second Paragraph: The last sentence does not make sense, change “human milk” to “nutrition”.

Data Collection:

- Explain what directed content analysis entails.

- When you say that a researcher and a participant review the relationships between themes and subthemes, do you mean that you checked along the way with differing participants or that you used one in particular? Clarify this. If only one participant was used, how did you pick which one? Did you discuss during the data collection or after? Etc.

Results

Overview:

- More detail is needed in text for your results. Although it is in your tables it is helpful to the reader if the important values are also in the text.

Sample Characteristics:

- Add in more data for the stats given. What was the SD for the age and monthly income? What is the percentage of the sample educated higher than secondary school, working in government or state, primipara and multipara? Include mean gestation.

- You need to specify that the diagnosis is for T2DM

- Give the SD and range for T-IFI score and BSES-SF

- How did you divide the participants into two groups for the BSES-SF, clarify. Eg. “Low-confidence group (<50) being… high-confidence group (>50) being…”

Qualitative Findings

Overview:

- Include numbers next to all your claims. For example: “Some participants (3/12) indicted that…”

- If only one participant expressed concern over their ability to feed their baby should that quote be used as your title? The title sets it up as if this is a dominant theme that was found. It also implies a sense of negativity, whereas the interview themes seem very positive. Instead, it may be better to use another quote from a theme that was more prominent such as the intention to breastfeed until they stop producing milk (if this is indeed a large percentage).

Theme 3:

- (Individual) This paragraph was difficult to read, rephrase if possible. Additionally, in the last sentence there is an error “that wasn’t was not enough”.

- (Societal) I am not sure if this would count as a societal level issue and not a personal? Unless it is expected of them by society to send away their infants? If this is a common societal practice/expected of mothers, please explain more in the background to help contextualize the reader.

- (Societal) Include how many of the mothers were planning on living in different provinces from their infants.

Theme 4:

- You say that each level has two subthemes but I am unable to identify what they are. Make them more explicit.

- (Individual) At the end you have an ID but there is no direct quote.

- (Individual) How are the benefits and bonding aspect of breastfeeding a facilitator? I think they should be their own thing separated from cost-effectiveness and immune boosting – which it is clear how those two are classed as facilitators.

- (Interpersonal level) in the final sentence it should either be “they planned to restrict their diet…” or “she planned to restrict her diet…”

- (Societal level) There are a lot of typos in this section.

- (Societal level) What do you mean by “alternatively” in this paragraph? Alternative to what?

Member Checking:

- This section has a lot of typos and grammatical mistakes.

- The final sentence does not make sense. How do you draw the conclusion that as affordability depends on the mother’s financial status that the key factor is mother’s desire? Surely this sentence shows that the key factor is income/job/financial status and not her desire?

Discussion

Overview

- Not much emphasis or discussion on the impact of T2DM particularly in your findings – you do not mention any T2DM findings until paragraph 4. You need to focus more on how T2DM interplays with the pregnant women’s breastfeeding intentions and confidence. Focus on T2DM first in your discussion as this should be the main focus of your paper. You can then move to more general recommendations later (eg, the need to extend maternity leave for all mothers).

- The discussion is very disjointed, as each paragraph addresses a finding without leading into the next. There needs to be more flow and tying the points together to make a larger overview rather than 9 separate points. This may be assisted when integrating diabetes into each point.

Paragraph 2:

- Did all participants express intention to BF until they exhausted their breast milk or just most/some?

Paragraph 3:

- You claim that these findings are similar to other studies that show that breastfeeding experience is a predictor for subsequent breastfeeding – however, as you interview women while they are pregnant you only measure breastfeeding intention and are unable to state if your participants actually breastfed or not. Be careful with phrasing in this regard.

- The final sentence is incomplete.

Paragraph 4:

- You discuss how participants were worried that BF would cause high blood sugar and were unsure if feeding BM would be harmful to their babies, but you do not mention this in your results so this is the first that the reader is discovering this. Add this finding into your results so you can address it better in the discussion.

- This paragraph should be a bigger emphasis in your discussion – as the impact of T2DM on BF is the main driver of your paper. Expand more on this content.

Paragraph 5:

- Same as previous paragraph, this key finding was not discussed in your results section. Be sure to add it in.

- This finding is very important! Be sure to emphasize it more.

Paragraph 6:

- This is the context we needed earlier to understand better the results on mother/infant separation. I would move the background information into the Setting and Relevant Context section.

Paragraphs 8 & 9:

- Are these foods safe/recommended for people with T2DM?

- These need to be tied into your specific sample of women with T2DM, rather than just general pregnancy as it is phrased now.

6. PLOS authors have the option to publish the peer review history of their article (what does this mean? ). If published, this will include your full peer review and any attached files.

**Do you want your identity to be public for this peer review?** For information about this choice, including consent withdrawal, please see our Privacy Policy .

Reviewer #1: No

Reviewer #2: No

---

## [Decision Letter · Decision Letter 1]

6 Jan 2025

"Feeding the baby breast milk shouldn’t be a problem" Breastfeeding confidence and intention in pregnant persons with type 2 diabetes mellitus from Thailand

PGPH-D-24-01048R1

Dear Dr. Phonyiam,

We are pleased to inform you that your manuscript '"Feeding the baby breast milk shouldn’t be a problem" Breastfeeding confidence and intention in pregnant persons with type 2 diabetes mellitus from Thailand' has been provisionally accepted for publication in PLOS Global Public Health.

Best regards,

Julia Robinson

Staff Editor

Reviewer Comments (if any, and for reference):

Reviewer's Responses to Questions

**Comments to the Author**

1. If the authors have adequately addressed your comments raised in a previous round of review and you feel that this manuscript is now acceptable for publication, you may indicate that here to bypass the “Comments to the Author” section, enter your conflict of interest statement in the “Confidential to Editor” section, and submit your "Accept" recommendation.

Reviewer #2: All comments have been addressed

2. Does this manuscript meet PLOS Global Public Health’s publication criteria ? Is the manuscript technically sound, and do the data support the conclusions? The manuscript must describe methodologically and ethically rigorous research with conclusions that are appropriately drawn based on the data presented.

Reviewer #2: Yes

3. Has the statistical analysis been performed appropriately and rigorously?

Reviewer #2: N/A

4. Have the authors made all data underlying the findings in their manuscript fully available (please refer to the Data Availability Statement at the start of the manuscript PDF file)?

Reviewer #2: Yes

5. Is the manuscript presented in an intelligible fashion and written in standard English?

Reviewer #2: Yes

6. Review Comments to the Author

Reviewer #2: (No Response)

7. PLOS authors have the option to publish the peer review history of their article (what does this mean? ). If published, this will include your full peer review and any attached files.

**Do you want your identity to be public for this peer review?** For information about this choice, including consent withdrawal, please see our Privacy Policy .

Reviewer #2: No
